# Synthesis of Calcium Silicate Hydrate from Coal Gangue for Cr(VI) and Cu(II) Removal from Aqueous Solution

**DOI:** 10.3390/molecules26206192

**Published:** 2021-10-14

**Authors:** Qing Zhang, Guijian Liu, Shuchuan Peng, Chuncai Zhou

**Affiliations:** 1School of Resource and Environmental Engineering, Hefei University of Technology, Hefei 230009, China; 2020010065@mail.hfut.edu.cn (Q.Z.); zhoucc@hfut.edu.cn (C.Z.); 2State Key Laboratory of Loess and Quaternary Geology, Institute of Earth Environment, The Chinese Academy of Sciences, Xi’an 710075, China; lgj@ustc.edu.cn; 3CAS Key Laboratory of Crust-Mantle Materials and Environment, School of Earth and Space Sciences, University of Science and Technology of China, Hefei 230026, China

**Keywords:** removal, Cr(VI), Cu(II), aqueous solution, coal gangue

## Abstract

Both the accumulation of coal gangue and potentially toxic elements in aqueous solution have caused biological damage to the surrounding ecosystem of the Huainan coal mining field. In this study, coal gangue was used to synthesize calcium silicate hydrate (C-S-H) to remove Cr(VI) and Cu(II)from aqueous solutions and aqueous solution. The optimum parameters for C-S-H synthesis were 700 °C for 1 h and a Ca/Si molar ratio of 1.0. Quantitative sorption analysis was done at variable temperature, C-S-H dosages, solution pH, initial concentrations of metals, and reaction time. The solution pH was precisely controlled by a pH meter. The adsorption temperature was controlled by a thermostatic gas bath oscillator. The error of solution temperature was controlled at ± 0.3, compared with the adsorption temperature. For Cr(VI) and Cu(II), the optimum initial concentration, temperature, and reaction time were 200 mg/L, 40 °C and 90 min, pH 2 and 0.1 g C-S-H for Cr(VI), pH 6 and 0.07 g C-S-H for Cu(II), respectively. The maximum adsorption capacities of Cr(VI) and Cu(II) were 68.03 and 70.42 mg·g^−1^, respectively. Furthermore, the concentrations of Cu(II) and Cr(VI) in aqueous solution could meet the surface water quality standards in China. The adsorption mechanism of Cu(II) and Cr(VI) onto C-S-H were reduction, electrostatic interaction, chelation interaction, and surface complexation. It was found that C-S-H is an environmentally friendly adsorbent for effective removal of metals from aqueous solution through different mechanisms.

## 1. Introduction

Owing to the excessive extraction of coal deposits in China’s Huainan, the aqueous solution area has been a unique part of the surface water bodies [1]. The aqueous solution area in Huainan reached nearly 1.493 × 10^4^ ha in 2017 [2]. As a result, the restoration of heavy metal–polluted aqueous solution is critical for utilizing and protecting water resources. The Cu(II) and Cr(VI) are two common metallic elements found in aqueous solution that have long-term harmful effects on humans and the ecosystem [3]. As compared with Cr(III), Cr(VI) is considered to be strongly carcinogenic to humans [4]. Similarly, excessive Cu(II) can also cause damage to human organs and tissues [5]. In China, surface water quality standards are 0.05 mg/L for Cr(VI) and 1.0 mg/L for Cu(II). So far, several technologies have been carried out to abate water pollution, including adsorption, photocatalysis, redox, chemical precipitation, membrane separation, and solvent extraction [6,7,8,9,10]. Among these techniques, adsorption has been considered a viable approach emerging as an environment-friendly option for heavy metals removal from contaminated matrices [11,12].

Besides this, annual coal output is over 100 million tons in Huainan city, which is one of the largest coalfields in China [13]. Coal gangue (CG) was discharged from coal separation and excavation, which accounted for about 10–15% of the annual coal output [14,15]. CG contains high amounts of Al_2_O_3_ and SiO_2_; therefore, it can be used as a cementing material [16]. Beyond that, CG also contains a certain amount of carbon, which can be used as source of energy [17]. CG also has high organic matter, nitrogen, and phosphorus; thus, it can improve soil porosity and nutrient status [18]. However, so far, little literature has reported the preparation of calcium silicate hydrate (C-S-H) from aluminum (Al) and silicon (Si)–abundant CG for the adsorption of metals from the perspective of experimental design, adsorption kinetics, and isotherm.

According to many reports, C-S-H has a porous structure with a great specific surface area, called a green adsorbent [19,20]. C-S-H has high adsorption efficiency for metal ions in the polluted environment. However, C-S-H applications are restricted because the traditional synthesis of C-S-H from cement costs a relatively high amount and also causes secondary pollution [21]. Previous studies have reported that waste materials such as coal ash, bottom ash, and sludge can supplement cementitious material for cement production [22,23]. One thing all these materials have in common is that they are pozzolanic by-products and rich in Al and Si, having cementitious potential, which can be reacted with lime to prepare C-S-H [24]. CG was also a pozzolanic by-product and rich in Al and Si with cementitious potential [16]. Consequently, it is of great interest to replace cement to prepare cost-effective and environmentally friendly C-S-H from CG.

The present study investigated the potential utilization of Al and Si–abundant CG as a raw material to prepare C-S-H and determine optimal parameters for producing C-S-H. This work then defined the adsorptive interaction of C-S-H based CG for removing Cr(VI) and Cu(II) from contaminated water, followed by adsorption experiments with different sorption parameters, e.g., initial solution pH. The overarching objectives were explored the adsorption kinetics and isotherm modeling. Moreover, the recycling and desorption experiments were assessed to explore the generated moieties. Finally, the adsorption potential of CG in a multi-metal-polluted environment was elucidated.

## 2. Results and Discussion

### 2.1. Characterization of CG

The diffraction pattern of CG is presented in Appendix A. The two mineral phases of CG were kaolinite (Al_2_O_3_·2SiO_2_·2H_2_O) and quartz (SiO_2_). Kaolinite was mainly composed of a Si-oxygen tetrahedron ([SiO_4_]^4−^) and aluminum-oxygen octahedron ([AlO_6_]^3−^). The sublayers of [SiO_4_]^4−^ and [AlO_6_]^3−^ were annular structures connected by oxygen atoms to form hexagonal closed rings. Previous studies have reported that OH^−^ groups in kaolinite could be gradually removed after calcination. As a result, [AlO_6_]^3−^ was transformed into Al-oxygen tetrahedron ([AlO_4_]^5−^), and kaolinite was transformed into metakaolin (Al_2_O_3_·2SiO_2_) [25]. As an amorphous material, metakaolin had high pozzolanic activity [26]. Moreover, the dissolution of Al and Si contained in CG could be accelerated in the alkaline activator [27]. Thus, Al and Si abundant CG could be used as a precursor to prepare C-S-H. The CG morphology was observed by SEM and are presented in Appendix A, where it gives the undulant and rough surface of CG. The image shows the exact inner mineral morphology with large size particles.

### 2.2. Optimization of the Synthesis Process

#### 2.2.1. Effect of Calcination Temperature onto C-S-H

C-S-H was synthesized by the following three chemical reactions [28]: (i) dissolution of CG and Ca(OH)_2_, in which the Si-O and Al-O bands are broken, and the aluminosilicate vitreous are dissociated; (ii) the monomer of Si and Al are reconstructed and reformed with a lower degree of polymerization; (iii) [SiO_4_]^4−^ and [AlO_4_]^5−^ are condensed into a three-dimensional netlike inorganic polymer. Ca(OH)_2_ and CG were involved in the vitreous dissolution and the spatial skeleton construction during the whole process. The calcination temperature was the key factor to the synthesis process, as Konan et al. [29] reported that CG had a decline of pozzolanic activity with a high or low calcination temperature. The calcination temperature effect was investigated under calcination time (1 h) and Ca/Si (1.0). Appendix A displays the XRD spectrum of the synthesized samples calcinated at different temperatures. It showed that the characteristic diffraction peak of C-S-H was sharp and intense at 700 °C. The synthesized samples were transformed to mullite and Si-spinel over 700 °C with low pozzolanic activities [30]. As a result, this work identified 700 °C as a suitable calcination temperature.

The TG-DTG curves of CG and C-S-H are exhibited in Appendix A. The TG-DTG curves of raw CG could be identified in three parts: (i) The first endothermic peak appeared from ambient to 100 °C, which resulted in slight weight loss because of the volatilization of adsorbed water. The losing weight kept a constant at 100–200 °C. (ii) The second endothermic peak appeared around 300–750 °C. The kaolinite lost its crystal water through destruction of the crystal structure. Meanwhile, the endothermic dehydroxylation occurred around 300 °C, which resulted in a sharp weight loss due to combustion of the fixed carbon and its subsequent volatilization escape; the maximum weight loss occurred at 470 °C. (iii) Due to mineral decomposition, a third endothermic peak occurred between 750 and 950 °C, and the residual mass was 82%. When compared to CG, the weight of C-S-H decreased slightly from ambient to 600 °C and was in a definite downward trend from 600 to 780 °C. The endothermic and exothermic peaks of C-S-H appeared at 775 °C and 790 °C, respectively. The glass phase structure with thermodynamic instability was formed, resulting in the enhanced C-S-H cementitious activity. Furthermore, the structure chains of [SiO_4_]^4−^ and [AlO_6_]^3−^ was destroyed. As a result, the tetrahedron of Si-O and trihedron of Al-O could not form long chains. Ca^2+^ would reselect their interstitial position. This result was in agreement with XRD patterns (Appendix A).

#### 2.2.2. Effect of Calcination Time onto C-S-H

The calcination time effect was investigated under the calcination temperature 700 °C and Ca/Si = 1.0. The XRD patterns (Appendix A) presented that the high crystallinity C-S-H could be detected after calcination for 1 h, indicating that the dihydroxylation of kaolin had been completed [31]. Then, C-S-H was formed when metakaolin was reacted with Ca(OH)_2_. The FTIR spectra (Appendix A) showed that CG revealed stretching vibrations of outer and inner OH^−^ groups at 3653.48 and 3620.70 cm^−1^, respectively. The 538.52 and 913.61 cm^−1^ bands were attributed to the stretching vibration of Si-O-Al^vin^ and Al-O-H, respectively; similarly, peaks at 1163.83 and 778.62 cm^−1^ originated since the existence of Si-O-Si groups. The stretching peaks at 1033.66, 695.69, and 471.51 cm^−1^ exhibited the presence of Si-O-bands. After calcination, the Al-O-H band, the Si-O-Al^va^ vibration, and the OH^−^ groups disappeared, indicating kaolinite transformation to metakaolin. At the same time, the 1095 cm^−1^ band was related to Si-O vibration. Bands at 877 and 798 cm^−1^ were linked to Si-O-Si vibration. It indicated the decomposition of SiO_2_ to form a new siliceous phase, which proved the formation of Si-O-Ca bands [32]. The outcome was similar to that of XRD patterns (Appendix A).

According to Guo et al. [33], the dissolution characteristic of Si^4+^ and Al^3+^ ion was related to the pozzolanic activity of CG and calcination time. Appendix A shows that the maximum dissolution amount of Si^4+^ and Al^3+^ ions from the C-S-H in NaOH solution was 180.73 mg·g^−1^ by the ICP test, indicating that the synthesized C-S-H had the highest pozzolanic activity after 1 h at 700 °C and fully decomposed to active Al_2_O_3_ and SiO_2_. Hence, this work defined a suitable calcination time of 1 h.

#### 2.2.3. Effect of Ca/Si Ratio onto C-S-H

Lothenbach and Nonat [34] reported that C-S-H and Si (amorphous) could appear simultaneously when the Ca/Si ratio was less than 0.67, whereas both Ca(OH)_2_ and C-S-H would coexist when the Ca/Si ratio was less than 0.67. Hence, the Ca/Si ratio had a major influence on C-S-H structural characteristics [35]. The experiments of the Ca/Si ratio were investigated at 700 °C and 1 h. As seen in Appendix A, the well and high peak of the Ca/Si ratio of 1.0 was detected without amorphous silica and Ca(OH)_2_ formed, indicating that high silica contents contributed to the Ca phase in a rapid reaction and was beneficial to form C-S-H. Kapeluszna et al. [36] also stated that the high purity C-S-H phase could be detected when the Ca/Si ratio was 1.0. Besides this, previous studies confirmed that Ca/Si ratio was proportional to Ca-OH bonds while inversely proportional to the concentration of Si-OH bonds [37]. The decreased Si-OH bonds were not conducive to form C-S-H. Similarly, the increased Ca-OH bonds were benefited the C-S-H formation.

As seen in Appendix A, it could find that C-S-H filled that space with the particles and connected fine-grained materials closely. The larger diameter and high C-S-H porosity could provide more reactive binding sites for adsorbing metal ions as mesoporous material [38]. The specific surface area of CG was 13.84 m^2^/g with a small porous structure (0.03 cm^3^/g). However, the specific surface area of C-S-H was 24.25 m^2^/g and had a larger porous structure of 0.13 cm^3^/g after calcination.

### 2.3. Cr(VI) and Cu(II) Adsorption onto C-S-H

#### 2.3.1. Effect of pH

Solution pH was a critical factor in exploring the adsorption process. In this work, the effect of pH was investigated in the pH range of 1 to 10. The C-S-H surface was positively charged when solution pH was less than the zero point of charge for C-S-H. Oppositely, the C-S-H surface was negatively charged when solution pH was higher (Figure 1a).

Fang et al. [39] stated that Cr_2_O_7_^2−^ and HCrO_4_^−^ were its main form at a pH range of 2–6, and CrO_4_^2−^ was its main form at pH > 6. Besides this, Cr_2_O_7_^2−^ was more preferably adsorbed than HCrO_4_^−^ due to that the charge/radius ratio of HCrO_4_^−^ was lower than that of Cr_2_O_7_^2−^ [40]. As seen in Figure 1b, at pH lower than 2, Cr_2_O_7_^2−^ and HCrO_4_^−^ were competing for the C-S-H surface sites, H^+^ also could destroy the internal chain structure of C-S-H [41], resulting in the decreased Cr(VI) removal efficiency. Negatively charged Cr(VI) (Cr_2_O_7_^2−^, HCrO_4_^−^, and CrO_4_^2−^) competed with protons for negatively charged binding sites on the C-S-H surface when the pH was between 4 and 10 [42]. The electrostatic attraction between C-S-H and Cr(VI) would gradually decrease and approach zero at pH 10. However, at pH lower than 2, Cu^2+^ positively charged competed with protons for the positively charged binding sites C-S-H surface (Figure 1c). Meanwhile, Cu(II) showed a strong electrostatic interaction with the C-S-H when the pH was between 4 and 6; however, Cu(II) would undergo hydrolysis at a pH higher than 6.0. The hydrolyzed Cu(II) species might be generated, so the Cu(II) adsorption measurements become difficult. Our results were in line with Gonçalves et al. [40] and Fang et al. [39].

#### 2.3.2. Effect of C-S-H Dosage and Initial Sorbent Concentration

The adsorbent dosages (0.05–0.15 g) and initial sorbent concentrations (25–1000 mg/L) were undertaken in this study. The result showed that maximum removal efficiency of Cr(VI) was reached with C-S-H dosages of 0.1 g, and Cu(II) was reached with C-S-H dosages of 0.07 g (Figure 2a). Further, Cr(VI) removal remained above 99% when its initial concentration less than 200 mg/L. Then, Cr(VI) removal began to decrease gently with the increased initial concentrations (200–1000 mg/L), reaching the lowest removal (68.32%) occurring at 1000 mg/L (Figure 2b). A similar trend was observed for Cu(II) removal, when the initial concentration of Cu(II) was 25 to 300 mg/L, the removal was more than 99%. However, as the concentration increased from 300 to 1000 mg/L, the removal was only 74.62%. It could be explained as metal ions anchored on the C-S-H surface and blocked porous structure, leading to the reduced exposed surface area during the adsorption process [37]. Thus, 0.1 g of C-S-H dosage was suitable to remove Cr(VI) with original Cr(VI) (25–200 mg/L) concentrations. Meanwhile, 0.07 g of C-S-H dosage was sufficient to eliminate Cu(II) with initial Cu(II) (25–300 mg/L) concentrations.

#### 2.3.3. Effect of Temperature

Figure 2c shows that the removal efficiency of Cr(VI) increased as the elevated temperature was raised from 25 °C to 40 °C, which presented the maximum removal at 40°C. The removal rate decreased significantly over 40 °C, suggesting the adsorption process was endothermic, then exothermic. This possible reason was that active sites in the C-S-H were deactivated with higher temperatures [43]. However, the Cu(II) removal increased with the temperature increase from 25 °C to 45 °C, suggesting the isomorphic substitution of C-S-H with Cu(II) [5].

#### 2.3.4. Adsorption Isotherm

Four isotherm models were selected to fit the results (Equations (1)–(4)). The Langmuir isotherm model assumed that the adsorbent surface was uniform and an utterly independent monolayer adsorption process for adsorbate. The Freundlich isotherm model described the polyphase adsorption equilibrium on the adsorbent surface. The Temkin isotherm model defined the chemisorption of adsorbents to adsorbate. In contrast, the D-R isotherm model was suitable for physisorption processes [44]. The experimental data was performed at 40 °C for 90 min, [Cr(VI)]_0_ = [Cu(II)]_0_ = 25–1000 mg/L, pH 2 and 0.1 g C-S-H for Cr(VI), pH 6 and 0.07 g C-S-H for Cu(II). The isotherm equation parameters are provided in Table 1.

Langmuir, Freundlich, Temkin and Dubinin–Radushkevich (D-R) isotherm models (Equations (1)–(4)) were carried out by the following [44,45]:(1)$$\frac{C_{e}}{q_{e}}=\frac{1}{K_{L}q_{m}}+\frac{C_{e}}{q_{m}}$$
(2)$$\log q_{e}=\log K_{F}+\frac{1}{n}\log C_{e}$$
(3)$$q_{e}=\frac{RT}{b}\ln K_{T}+\frac{RT}{b}\ln C_{e}$$
(4)$$\ln q_{e}=\ln q_{d}-\beta \varepsilon^{2}$$
where C_e_ (mg·L^−1^) is the adsorbate equilibrium concentration in solution, q_e_ (mg·g^−1^) is the adsorbate equilibrium mass, q_m_ (mg·g^−1^) is the adsorbate maximum mass. K_L_ (L·mg^−1^) is the Langmuir isotherm model constant. Meanwhile, K_F_ (L·mg^−1^) denotes the Freundlich isotherm model constant and is related to the adsorption capacity, while n is the constant to describe the adsorption strength. If the n value is greater than 1, it not only indicates the existence of chemical adsorption, but also suggests the high affinity between the adsorbate and the adsorbent. On the other hand, b (J·mol^−1^) is the constant related to adsorption heat; K_T_ (L·g^−1^) is the Temkin isotherm model constant; R (8.314 J·mol^−1^·K^−1^) is the perfect gas constant and *T* (K) is the solution temperature; q_d_ (mg·L^−1^) is the D-R model constant; β is the constant related to adsorption free energy; and ε is the adsorption potential.

The greater R^2^ for Langmuir isotherm showed that both Cr(VI) and Cu(II) adsorption onto C-S-H were monolayers [39]. The largest adsorption capacity of Cr(VI) was 68.03 mg·g^−1^ and Cu(II) was 70.42 mg·g^−1^. Table 2 presents that C-S-H had better adsorption performance than some previously reported adsorbents.

#### 2.3.5. Adsorption Kinetics and Intra-Particle Diffusion Analysis

Figure 3a indicates that both Cu(II) and Cr(VI) showed three distinct regions in the whole adsorption process by the intra-particle diffusion model (Equation (5)). For Cu(II), the first region was from 1 to 25 min, the second was from 25 to 90 min, and the third was from 90 to 180 min. For Cr(VI), the first region appeared between 1 to 30 min, the second between 30 to 90 min, and the third approached within 90 to 180 min. The particle diffusion and surface diffusion were the adsorption rate-limiting steps. Additionally, the slope of the first and second stages represented the rate of surface diffusion and particle diffusion, respectively [5]. Meanwhile, the larger the intercept of the second region, the higher the effect of surface diffusion. The third region reached adsorption equilibrium. The results showed that the rate of surface diffusion was higher than particle diffusion. Furthermore, particle diffusion played a significant role in increasing temperature, which also presented the tremendous and quick adsorption potential of C-S-H for Cu(II) and Cr(VI).

Intra-particle diffusion model was expressed as follows (Equation (5)) [44,45]:(5)$$q_{t}=k_{i}t^{0.5}+C$$
where k_i_ (g·mg^−1^·h^−1^) is the intra-particle diffusion rate constant while constant C depends on the thickness of the boundary layer.

The pseudo-first order kinetic model (Equation (6)) and pseudo-second order kinetic model (Equation (7)) were used as follows [45]:(6)$$\frac{dq_{t}}{dt}=k_{1}(q_{e}-q_{t})$$
(7)$$\frac{dq_{t}}{dt}=k_{2}{\left(q_{e}-q_{t}\right)}^{2}$$
where q_t_ and q_e_ (mg·g^−1^) denote the amounts of adsorbate at time t (min) and at equilibrium, respectively. Similarly, k_1_ (min^−1^) and k_2_ (g·mg^−1^·min^−1^) present the pseudo-first rate constant and the pseudo-second rate constant, respectively.

The pseudo-first-order and pseudo-second-order (Equations (6) and (7)) kinetic models were chosen to fit the experimental data. The adsorption kinetics and intra-particle diffusion analysis were carried out at [Cr(VI)]_0_ = [Cu(II)]_0_ = 200 mg/L, 40 °C, reaction time 1–180 min, pH 2 and 0.1 g C-S-H for Cr(VI), pH 6 and 0.07 g C-S-H for Cu(II). As shown in Figure 3b,c, the experimental results were similar to pseudo-second-order kinetic model with a greater R_2_^2^ (>0.99%). The pseudo second rate constant (k_2_) of Cu(II) and Cr(VI) were 10.02 and 13.50 g·mg^−1^·min^−1^, respectively.

#### 2.3.6. Adsorption Thermodynamics

The thermodynamic adsorption parameters were calculated utilizing the Langmuir equilibrium constant (Appendix A). The values of Gibbs free energy change (ΔG^0^) were negative, denoting that the transfer processes of Cu(II) and Cr(VI) were feasible and spontaneous from an aqueous solution to the C-S-H surface. The enthalpy change values (ΔH^0^) and entropy change (ΔS^0^) were positive, suggesting that the adsorption was an endothermic process and entropically favorable. Beyond that, the physisorption and chemisorption of Cu(II) and Cr(VI) existed simultaneously, where physisorption showed dominance.

The enthalpy change (ΔH^0^, J·mol^−1^), Gibbs free energy change (ΔG^0^, J·mol^−1^), and entropy change (ΔS^0^, J·mol^−1^·K^−1^) and activation energy (E_a_) were used as follows (Equations (8)–(11)) [44]:(8)$$k_{2}=A\exp\left(-\frac{E_{a}}{RT}\right)$$
(9)$$\Delta G^{0}=-RT\ln K_{d}$$
(10)$$\ln K_{d}=\frac{\Delta S^{0}}{R}-\frac{\Delta H^{0}}{RT}$$
(11)$$K_{d}=\frac{q_{e}}{C_{e}}$$

#### 2.3.7. Recycling and Regeneration Performance

The experimental data were carried out at [Cr(VI)]_0_ = [Cu(II)]_0_ = 200 mg/L, 40 °C, 90 min, pH 2 and 0.1 g C-S-H for Cr(VI), pH 6 and 0.07 g C-S-H for Cu(II). The removal efficiency of Cu(II) and Cr(VI) gradually dropped with the increased number of regeneration cycles (Appendix A). These results showed that C-S-H was stable for multi-time usage despite six adsorption/desorption cycles.

#### 2.3.8. Cu(II) and Cr(VI) Removal from Aqueous Solution

C-S-H confirmed that it could effectively remove Cu(II) and Cr(VI) from aqueous solution. After adsorption by C-S-H, the contents of Cu(II) and Cr(VI) from the aqueous solution reduced from 1.64 and 0.083 mg/L to 0.17 and 0.022 mg/L, respectively. The Cu(II) and Cr(VI) removal efficiency were 89.63% and 73.49%, respectively. Although the pH (5.5) of aqueous solution was not conducive to Cr(VI) removal, it still met the surface water quality standards in China (0.05 mg/L). The solution pH increased from 5.5 to 5.8 after the reaction. C-S-H was an alkaline material. Adding a small amount of C-S-H into the solution could cause a slight increase in pH value. The efficiency of C-S-H to absorb other contaminants from aqueous solution will be studied in the future.

#### 2.3.9. Adsorption Mechanism

The XPS spectra of C-S-H after adsorption of Cu(II) and Cr(VI) were explored (Figure 4), and the results showed new peaks at 574.1 and 933.2 eV, indicating the Cr 2p and Cu 2p, respectively (Figure 4a). Moreover, the Cr 2p spectrum with high resolution was separated into two peaks at 587.3 eV (Cr 2p1/2) and 577.6 eV (Cr 2p3/2) (Figure 4b), suggested the existence of Cr(VI) and Cr(III). These results meant that Cr(VI) was reduced to Cr(III) by C-S-H. Besides this, Si-OH bands on the C-S-H surface could be chelation with Cr(III) at pH 2. Negatively charged Cr(VI) (Cr_2_O_7_^2−^ and HCrO_4_^−^) had a robust electrostatic interaction with the C-S-H at pH 2. The Cu 2p spectrum was separated into two peaks at 953.5 eV (Cu 2p1/2) and 933.2 eV (Cu 2p3/2) (Figure 4c), suggesting the feasibility of forming Cu(II) complexes, such as Si-OH-Cu^2+^. The possible proposed adsorption mechanism is presented in Figure 5.

## 3. Materials and Methods

### 3.1. Study Area and Raw Materials

CG sample and aqueous solution were brought from the Zhangji coalfield in Huainan, Anhui Province, China. Zhangji coalfield covers approximately 71 km^2^ with a total mining thickness of 21.08 m. The annual production capacity of the mine is 12.4 million tons. There are four representative aqueous solution bodies in the Zhangji coal mine, about 20 km^2^. Zhangji coal mine is one of the leading mines in Huainan and the first ten million tons modern mine in Anhui Province.

The raw CG was mainly composed of Al_2_O_3_ and SiO_2_ (Table 3). The pH value of aqueous solution was 5.5. The Cr(VI) and Cu(II) concentrations were 0.083 and 1.64 mg/L, respectively, which were the critical metal contaminants in aqueous solution. Ca(OH)_2_, NaOH, K_2_Cr_2_O_7_, Cu(NO_3_)_2_·3H_2_O, and HNO_3_ were of analytical grade and had not been purified further.

### 3.2. Synthesis of C-S-H

First, the CG was crushed by a high-speed pulverizer, ground, dried, passed through the 100-mesh sieve, and finally placed in a sealed bag for later use. Ca(OH)_2_ was added to CG to make the Ca/Si ratio of 0.6–1.5 as reported by the procedure expressed by Gineys et al. [55]. Then, a quick stirrer was used to mix CG and Ca(OH)_2_ thoroughly, and the composite samples were calcinated in a Muffle furnace at different temperatures (500, 600, 700, 800, and 900 °C) with various time (1, 2, 4, 6, and 8 h). After cooling, the composite sample was cured for 3 d at ambient temperature. The composite sample (1 g) was reacted with 1 mol/L 100 mL NaOH at 40 °C for 3 h to evaluate pozzolanic activity [33]. The powdered phase was the original phase of the produced C-S-H, which was used for the adsorption experiments to determine the adsorption capacity. The grain size of C-S-H was 0.15 mm.

### 3.3. Characterization

The specific surface area and pore diameter were examined with a specific surface area analyzer (BET, Waltham, MA, USA). The sample was observed for the microscopic morphology under the scanning electron microscope (SEM, Waltham, MA, USA). The chemical content of CG was determined by X-ray fluorescence spectrometry (XRF, Heidenheim, Germany). The thermal gravimetry (TG) was performed using a Shimadzu DTG-60H instrument (Shanghai, China) under a temperature range from ambient to 1000 °C at a constant rate (10 °C/min) under nitrogen conditions.

Fourier transform infrared spectroscopy (FTIR, Nicolet 8700, Florida City, FL, USA) and X-ray powder diffraction (XRD, TTR-VI, Heidenheim, Germany) were used to measure the structural and mineralogical composition of the sample. The XRD patterns of samples were stepped scanned at 2θ between 10° and 70° (λ = 1.54 Ǻ). The time per step was 0.15 s, and the step size was 0.02° 2θ. The samples were scanned from 400 to 4000 cm^−1^ by FITR.

The metal concentrations were measured using inductively coupled plasma mass spectrometry (ICP-MS, Optima 7300 DV, Johns Creek, GA, USA). Zeta potential was performed on the Zetasizer ZS instrument (Nanotrac Wave II, Johns Creek, GA, USA), and the test was repeated thrice for each sample. X-ray photoelectron spectrum (XPS, KRATOS, AXIS SUPRA+) was used for the valence states of surface elements of C-S-H.

### 3.4. Batch Adsorption Experiments

The metals adsorption experiments were carried out with a thermostatic gas bath oscillator at 200 rpm/min. K_2_Cr_2_O_7_ and Cu(NO_3_)_2_·3H_2_O were dissolved in deionized water to make Cr(VI) and Cu(II) aqueous solutions (25–1000 mg/L), respectively. Next, 10 mL of the aqueous solution was shaken with C-S-H in a 15 mL polyethylene tube. Adsorption experiments were performed with various adsorbent (C-S-H) dosages (0.05–0.15 g), pH (1–10), temperature (25–45 °C), initial concentration (25–1000 mg/L) of metals, and reaction time (1–180 min). Solution pH was adjusted by adding an acid (HNO_3_) or base (NaOH) solution. The pH value was kept constant during the experiment by a pH meter. Compared with the adsorption temperature, the error of solution temperature was controlled at ±0.3. After the oscillation, the supernatant solutions of all mixtures were collected by using a 0.45 μm membrane filter.

The 100 mL aqueous solution was shaken with C-S-H (1.0 g) in a 250 mL conical flask at 40 °C for 90 min. After agitation, the Cr(VI) and Cu(II) concentrations were tested using ICP-MS (Optima 7300 DV, USA).

The adsorption capacity (q_t_, mg·g^−1^) and removal efficiency (η, %) of Cr(VI) and Cu(II) were determined as follows (Equations (12) and (13)) [20]:(12)$$q_{t}=(C_{0}-C_{t})/m$$
(13)$$\eta =((C_{0}-C_{t})/C_{0})\times 100$$
where C_0_ and C_t_ (mg·L^−1^) represent the metal concentrations at the beginning of the experiment and at time t (min), respectively, and m (g) represents the adsorbent mass in a 1 L solution.

### 3.5. Recycling and Regeneration Performance

To know the efficacy of the adsorbent, 0.1 g of C-S-H was used by 0.1 M HNO_3_ solution for metal desorption. After the first oscillation, the adsorption residues were rinsed with water and then soaked in ethanol. Then, the same condition as the first test was performed for the repeated six adsorption–desorption cycles.

## 4. Conclusions

In our work, calcium silicate hydrate (C-S-H) was successfully prepared from coal gangue and applied to remove Cu(II) and Cr(VI) from aqueous solutions and aqueous solution. These results revealed that solution pH had substantial impact on adsorption and subsequent removal of metals from aqueous solution. Both Cu(II) and Cr(VI) followed the pseudo-second-order kinetic and Langmuir isotherm models, presenting the highest adsorption capacity of 70.42 and 68.03 mg·g^−1^, respectively. The particle diffusion was the primary adsorption rate limiting step by the intra-particle diffusion model. C-S-H was still stable after six cycles of adsorption and desorption. The removal mechanisms were: reduction of Cr(VI) to Cr(III), chelation of Si-OH with Cr(III), electrostatic interaction between Cr(VI) (Cr_2_O_7_^2−^ and HCrO_4_^−^) and C-S-H, and surface complexation (Cu(II) complexes). Future studies should concentrate on the commercial-scale applications of C-S-H for Cu(II) and Cr(VI) removal from aqueous solution while keeping an eye on other metals.

## Figures and Tables

**Figure 1 molecules-26-06192-f001:**
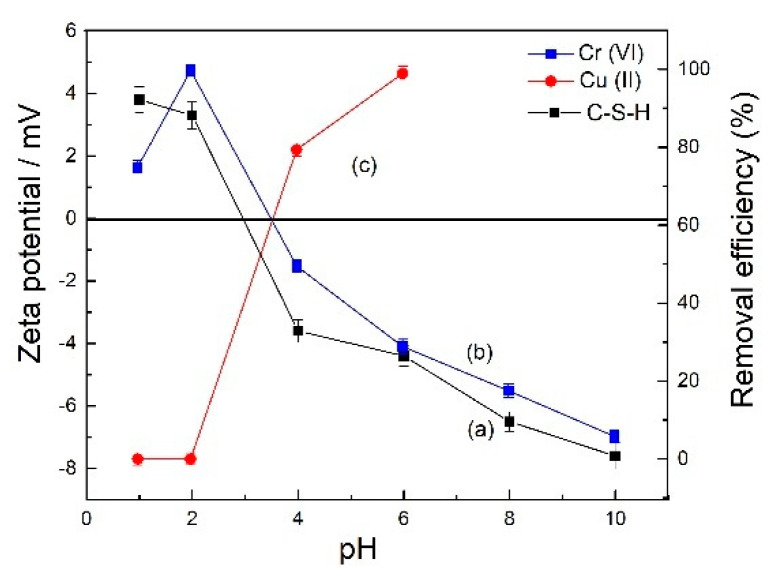
The zeta potential of C-S-H (**a**) and effect of pH for Cr(VI) (**b**) and Cu(II) (**c**) adsorption onto C-S-H (40 °C, [Cr(VI)]_0_ = [Cu(II)]_0_ = 200 mg/L, reaction time 90 min, 0.1 g C-S-H for Cr(VI), 0.07 g C-S-H for Cu(II)).

**Figure 2 molecules-26-06192-f002:**
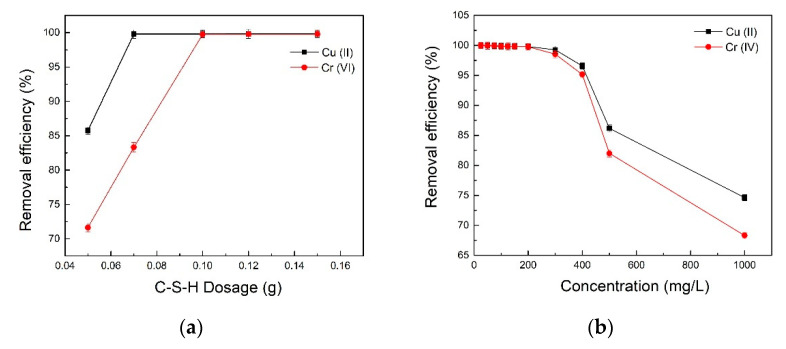
Effect of C-S-H dosage, initial sorbent concentration and temperature onto C-S-H ((**a**), 90 min, 40 °C, [Cr(VI)]_0_ = [Cu(II)]_0_ = 200 mg/L, pH 2 for Cr(VI), pH 6 for Cu(II)); (**b**), 40 °C, 90 min, pH 2 and 0.1 g C-S-H for Cr(VI), pH 6 and 0.07 g C-S-H for Cu(II); (**c**), [Cr(VI)]_0_ = [Cu(II)]_0_ = 200 mg/L, 90 min, pH 2 and 0.1 g C-S-H for Cr(VI), pH 6 and 0.07 g C-S-H for Cu(II)).

**Figure 3 molecules-26-06192-f003:**
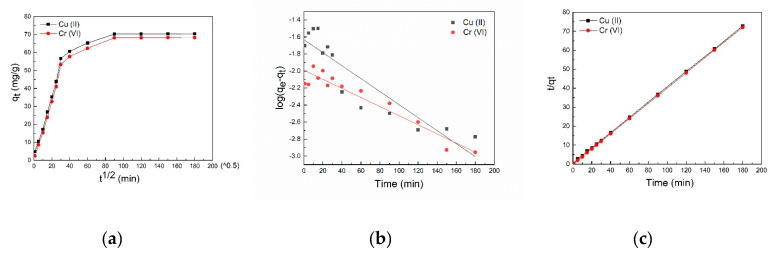
Intra-particle diffusion (**a**), pseudo-first-order (**b**), and pseudo-second-order kinetic models (**c**) of Cu(II) and Cr(VI) onto C-S-H.

**Figure 4 molecules-26-06192-f004:**
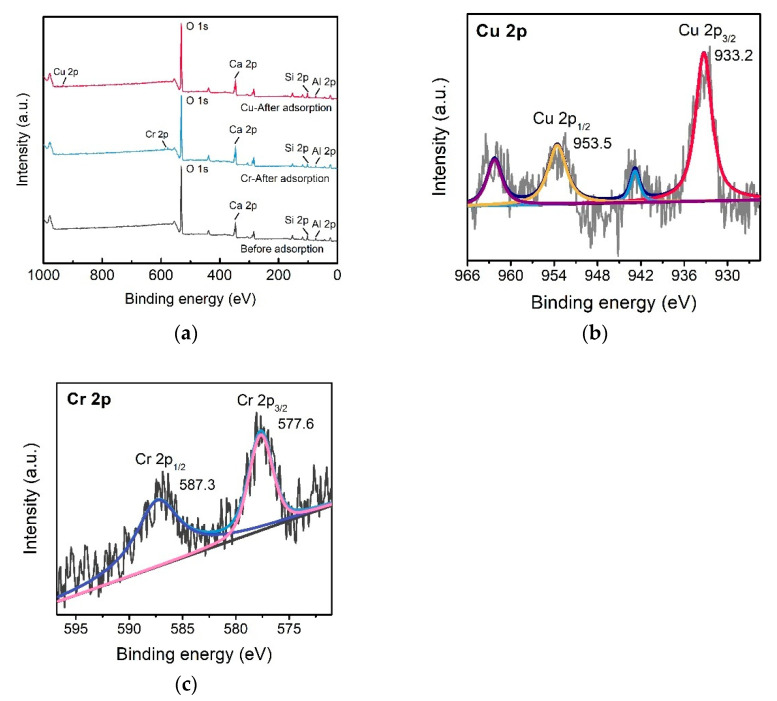
The XPS spectra of (**a**) C-S-H before and after adsorption, (**b**) Cu 2p of C-S-H after the reaction with Cu(II), and (**c**) Cr 2p of C-S-H after the reaction with Cr(VI).

**Figure 5 molecules-26-06192-f005:**
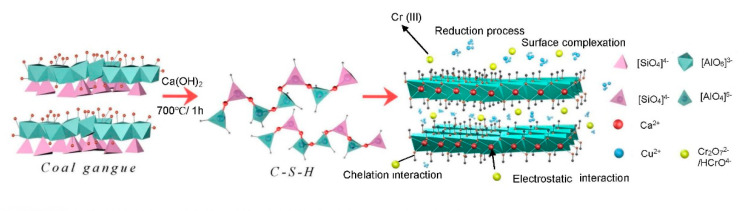
Synthesis of C-S-H and adsorption mechanism.

**Table 1 molecules-26-06192-t001:** The adsorption isotherms parameters for Cr(VI) and Cu(II) adsorption onto C-S-H.

Adsorbate	Isotherm	Parameters		R^2^
Cu(II)	Freundlich	K_F_ (L·mg^−1^)	n	0.79
		17.41	3.85	
	Langmuir	q_m_ (mg·g^−1^)	K_L_ (L·mg^−1^)	0.99
		70.42	0.15	
	D-R	q_d_ (mg·L^−1^)	β	0.93
		36.25	0.00	
	Temkin	K_T_ (L·g^−1^)	b (J·mol^−1^)	0.86
		4.48	435.81	
Cr(VI)	Freundlich	K_F_ (L·mg^−1^)	n	0.72
		16.41	3.86	
	Langmuir	q_m_ (mg·g^−1^)	K_L_ (L·mg^−1^)	0.98
		68.03	0.18	
	D-R	q_d_ (mg·L^−1^)	β	0.93
		33.29	0.00	
	Temkin	K_T_ (L·g^−1^)	b (J·mol^−1^)	0.92
		4.32	471.84	

**Table 2 molecules-26-06192-t002:** Comparison of Cr(VI) and Cu(II) adsorption results of C-S-H with other adsorbents.

Adsorbent	Metal Ion	q_m_ (mg/g)	Temperature (°C)	pH	Reference
Ca-Al-Zn-CO_3_ HTlc	Cu(II)	15.4	25	6	[46]
Black wattle tannin-immobilized nanocellulose		0.73	30	6	[47]
Carbon Prepared from Henna Leaves		0.057	35	6	[48]
Li-Al hydrotalcite-like compound		41.10	25	5	[5]
Grape bagasse		37.17	25	5	[49]
Opuntia biomass	Cr(VI)	16.5	20	2	[50]
Bismuth modified biochar		12.23	25	2	[51]
Dried Salvinia auriculata biomass		24.8	30	2	[52]
Barks of Acacia albida		2.98	37	2	[53]
Oedogonium hatei biomass		31	45	2	[54]
Rice husk		6.88	50	2	[43]
Fe (II)-rice husk		11.14	50	2	[43]
C-S-H	Cu(II)	70.42	40	6	Our work
	Cr(VI)	68.03	40	2	

**Table 3 molecules-26-06192-t003:** Chemical composition (wt%) of CG.

Type	Al_2_O_3_	SiO_2_	Fe_2_O_3_	TiO_2_	SO_3_	K_2_O	MgO	CaO	LOI
CG	32.31	51.60	2.57	0.91	0.66	1.81	0.54	0.55	9.05

## Data Availability

The data presented in this study are available in article.

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
