# Peer review of "Synthesis of Calcium Silicate Hydrate from Coal Gangue for Cr(VI) and Cu(II) Removal from Aqueous Solution"

_molecules, 2021, doi:10.3390/molecules26206192_

Round 1

Reviewer 1 Report

GENERAL IMPRESSION

This paper effectively demonstrated the synthesis of C-S-H from coal gangue and suggested its characteristics for the adsorption of Cu(II) and Cr(VI). Various results of experiments were clearly presented. Although the adsorption capacity was high, the temperature was too high and the pH was too low (for Cr(VI)) to be generalized. Please refer to the following specific comments.

GENERAL COMMENTS

  1. Although the resulting qm for Cr(VI) adsorption was very high (68.03 mg g-1), the adsorption isotherm experiment was conducted at pH 2, which is too low considering the possibly higher pH resulting from C-S-H. The actual qm at higher pH (7–10) should be also estimated or discussed regarding the result of the pH adsorption edge experiment. Also, the temperature (40 °C) seems to be too ideal. I suggest adsorption capacities at 15–25 °C are estimated or discussed.
  2. Was the synthesized C-S-H in the form of solid grain during the adsorption experiments? If so, the grain size of the resulting solid should be indicated. To apply as an adsorbent, discussion about the formation of the granular phase rather than the powdered phase is recommended.
  3. After experiments, pH may be changed due to C-S-H. The characteristics of C-S-H about pH change seem to be needed.
  4. The composition of the subsidence water is needed to consider the effect of ions other than Cu and Cr.
  5. Grammar should be checked and revised throughout the manuscript (e.g., Line 51, 135, 161, 182, 264). Spacing between figure and unit should also be revised (e.g., Line 102, 136). Also, language editing can be considered (e.g., Line 235).
  6. In the future, I suggest considering steel slag as well, because it has C-S-H component without consuming Ca(OH)2 which generates CO2 during production.

SPECIFIC COMMENTS

  1. Title: As ‘aqueous solution’ and ‘subsidence water’ can be overlapped partly, the terms should be modified throughout the manuscript including the title.
  2. Lines 34–35: As dissolved Cu is not so mobile in the environment, ‘far-reaching influence’ is not so adequate expression. Moreover, consistent use of Cu(II) and Cr(VI) throughout the manuscript is recommended.
  3. Lines 45–46: Isn’t coal gangue generated from excavation and separation rather than washing?
  4. Line 47: Is the spontaneous combustion of CG important regarding the research topic?
  5. Line 54: I think the term “litter” literature should be checked and revised.
  6. Line 71: As “highly crystalline” property is not generally needed to make an effective adsorbent, I suggest revising this sentence.
  7. Line 86–87: “the gas comprehensive control experimental base” is difficult to be understood. Revision and/or further explanation are required.
  8. Line 139–142: Substitution for the word “know” and “used” is recommended. The term “metal residues” also does not seem to coincide with the preceding explanation. What do the “repeated six” adsorptions mean? I think they should be specified as six adsorption-desorption cycles.
  9. Lines 178–181: Those sentences should be rewritten to explain exact phenomena systematically (e.g., 1) losing crystal water from kaolinite, 2) carbon combustion and escape of volatiles).
  10. Line 344–345: What was the pH after the reaction? Was it increased?

Reviewer 2 Report

The paper requires major revisions to be considered for publication afterwards. Firstly, it is necessary to check the text by an English language lecturer because many parts are poorly understood due to poor English. Then, there are claims in the paper that must be supported by appropriate references or additional experiments.

In the Abstract, it is claimed that the optimal initial concentration is 200 mg/L and that the adsorbent capacity for this concentration is 68 mg Cr(VI)/g (l. 22). However, at p. 6 (chapter 3.3.2) it was found that under the above conditions, capacity of the adsorbent is 20 mg/g.

Also, the authors did not explain why they examined the range of very high initial concentrations of the toxic ions (25-1000 mg/L).

According to experimental data the authors claimed that the adsorption of Cr(VI) is endothermic process up to 40 C and then, exothermic and did not offer any explanation. It should be explained. At p.9 (l. 330-332) they concluded that the adsorption is endothermic process. These different claims require the explanation.

At p. 4 the authors describe the recycling experiment. Why they soaked the adsorbent in ethanol after desorption and then rinsed it with water? The order is not logical. The data on the recycling experiment should be given in the text.

At p.8, the Table 3 is given but not discussed in the text. If the intention of the authors was to compare their results with that of different adsorbents, it is necessary that the comparison be based on the data obtained under similar experimental conditions.

At. p. 8 Table 2 is given. All parameters listed in the Table should be explained in the text.

At p. 9, the authors gave results of adsorption kinetics and intra-particle diffusion analysis. It is necessary to list the equations used for data processing in the text. Also, the graphs should be presented in an identical way.

At p. 10 the authors discussed the possible adsorption mechanism. The suggested mechanism is speculative because it is not based on experimental data. The reference (57) given as a conformation is quite inappropriate because in the ref. 57 a complex between Cr(III) and adsorbent is formed via Cr-N bonds.

Finally, when processing the text, the authors must pay attention to the fact that the units and names of the ions are stated uniformly. Also at Fig. S1 diffraction pattern is shown instead of “diffraction peak”. Two mineral phases can be seen instead of “phase structure”. At. p. 2 continuous stability (l.59) is mentioned. This is unclear and should be explained.

Round 2

Reviewer 1 Report

The manuscript was significantly improved following the previous comments. Please refer to the additional comments below.

  1. For the maximum adsorption capacities, as I think the conditions of temperature and pH are apart from the real conditions, the conditions should be specified in the manuscript including Abstract.
  2. The grain size of 0.15 mm should be indicated in the paper although it is small.
  3. The fact that the pH value was kept constant during the experiment should be added in the methods section.
  4. If the composition (other than Cr and Cu) of the subsidence water is available, please consider including it in the paper.
  5. Please check and remove the space between Cu and (II) and between Cr and (VI) throughout the manuscript.
  6. Line 381: a spacing is missing in “adsorption residues”
  7. The fact that pH increased from 5.5 to 5.8 during the reaction should be specified. Please refer to the previous responses to comments.

Reviewer 2 Report

The revised manuscrpt can be accepted for the publication.

Author Response

We are thankful to the reviewer for his comments to improve the quality of the manuscript. We are grateful to the reviewer for his approval of the revised manuscript.